# Deep Archimedean Copulas

**Chun Kai Ling**
Computer Science Dept.
Carnegie Mellon University
chunkail@cs.cmu.edu

**Fei Fang**
Institute for Software Research
Carnegie Mellon University
feif@cmu.edu

**J. Zico Kolter**
Computer Science Dept.
Carnegie Mellon University
Bosch Center for AI
zkolter@cs.cmu.edu

## Abstract

A central problem in machine learning and statistics is to model joint densities of random variables from data. Copulas are joint cumulative distribution functions with uniform marginal distributions and are used to capture interdependencies in isolation from marginals. Copulas are widely used within statistics, but have not gained traction in the context of modern deep learning. In this paper, we introduce ACNet, a novel differentiable neural network architecture that enforces structural properties and enables one to learn an important class of copulas–Archimedean Copulas. Unlike Generative Adversarial Networks, Variational Autoencoders, or Normalizing Flow methods, which learn either densities or the generative process directly, ACNet learns a generator of the copula, which implicitly defines the cumulative distribution function of a joint distribution. We give a probabilistic interpretation of the network parameters of ACNet and use this to derive a simple but efficient sampling algorithm for the learned copula. Our experiments show that ACNet is able to both approximate common Archimedean Copulas and generate new copulas which may provide better fits to data.

## 1 Introduction

Modeling dependencies between random variables is a central problem in machine learning and statistics. Copulas are a special class of cumulative density functions which specify the dependencies between random variables without any restriction on their marginals. This has led to long lines of research in modeling and learning copulas [15, 7], as well as their applications in fields such as finance and healthcare [5, 3]. Amongst the most common class of copulas are *Archimedean Copulas*, which are defined by a one-dimensional function $\varphi$, known as the generator, and often favored for their simplicity and ability to model extreme distributions. A key problem in the application of Archimedean Copulas is the selection of the parametric form of $\varphi$ as well as the limitations on the expressiveness of commonly used copula. Present workarounds include the selection of the best model between a fixed set of commonly used copulas, the use of methods based on information criterion such as Akaike and Bayesian Information Criterion (AIC, BIC), as well as more modern nonparametric methods.

In this paper, we propose ACNet, a novel network architecture which models the *generator* of an Archimedean copula using a deep neural network, allowing for network parameters to be learnt using backpropagation and gradient descent. The core idea behind ACNet is to model the generator as a sum of convex combination of a finite set of exponential functions with varying rates of decay, while exploiting their invariance to convex combinations and multiplications with other exponentials. ACNet is built from simple, differentiable building blocks, ensuring that log-likelihood is a differentiable function of $\varphi$, ensuring ease of training via backpropagation. By possessing a larger set of parameters, ACNet is able to approximate all copulas with completely monotone generators, a large class which encompasses most of the commonly used copulas, but also other Archimedean

copulas which have no straightforward closed forms. To our knowledge, ACNet is the first method to utilize deep representations to model generators for Archimedean copulas directly.

ACNet enjoys several theoretical properties, such as a simple interpretation of network weights in terms of a Markov reward process, resulting in a numerically stable, dimension independent method of sampling from the copula. Using this interpretation, we show that deep variants of ACNet are theoretically able to represent generators which shallow nets may not. By modeling the cumulative density directly, ACNet is able to provide a wide range of probabilistic quantities such as conditional densities and distributions using a *single* trained model. This flexibility in expression extends to both inference and training and is not possible with other deep methods such as Generative Adversarial Networks (GANs) or Normalizing Flows, which at best allow for the evaluation of densities.

Empirical results show that ACNet is able to learn standard copula with little to no hyperparameter tuning. When tested on real-world data, we observed that ACNet was able to learn new generators which are a better qualitative description of observed data compared to commonly used Archimedean copulas. Lastly, we demonstrate the effectiveness of ACNet in situations where measurements are uncertain within known boundaries. This task is challenging for methods which learn densities as evaluating probabilities would then involve the costly numerical integration of densities.

We (i) propose ACNet, the first network to learn completely monotone generating functions for the purpose of learning copulas, (ii) study the theoretical properties of ACNet, including a simple interpretation of network weights and an efficient sampling process, (iii) show how ACNet may be used to compute probabilistic quantities beyond log-likelihood and cumulative densities, and (iv) evaluate ACNet on both synthetic and real-world data, demonstrating that ACNet combines the ease of use enjoyed by commonly used copulas and the representational capacity of Archimedean copulas. The source code for this paper may be found at `https://github.com/lingchunkai/ACNet`.

## 2 CDFs and Copulas

Consider a $d$-dimensional continuous random vector $X = \{X_1, X_2, \cdots X_d\}$ with marginals $F_i(x_i) = \mathbb{P}(X_i \leq x_i)$. Given a $x \in \mathbb{R}^d$, the *distribution* function $F(x) = \mathbb{P}(X_1 \leq x_1, \cdots X_d \leq x_d)$ specifies all marginal distributions $F_i(x_i)$ as well as any dependencies between $X$. This paper focuses on continuous distribution functions which have well-defined densities.

### 2.1 Copulas

Of particular interest is a special type of distribution function known as a *copula*. Informally, copulas are distribution functions with uniform marginals in $[0, 1]$. Formally, $C(u_1, \cdots, u_d) : [0, 1]^d \to [0, 1]$ is a copula if the following 3 conditions are satisfied.

- (Grounded) It is equal to 0 if any of its arguments are 0, i.e., $C(\ldots, 0, \ldots) = 0$.
- It is equal to $u_i$ if all other arguments 1, i.e., for all $i \in [d]$, $C(1, \cdots, 1, u_i, 1, \cdots, 1) = u_i$.
- ($d$-increasing) For all $u = (u_1, \ldots, u_d)$ and $v = (v_1, \ldots, v_d)$ where $u_i < v_i$ for all $i \in [d]$,

$$\sum_{(w_1, \ldots w_d) \in \times_{i=1}^d \{u_i, v_i\}} (-1)^{|i:w_i = u_i|} C(w_1, \ldots, w_d) \geq 0. \tag{1}$$

Heuristically, the $d$-increasing property states that the probability assigned to any non-negative $d$-dimensional rectangle is non-negative.

Observe that the first 2 conditions are stronger than the limiting conditions required for distribution functions—in fact, groundedness coupled with the $d$-increasing property sufficiently define any distribution function. In particular, the second condition implies that Copulas have uniform marginals and hence, are special cases of distribution functions. Copulas have found numerous real world applications in engineering, medicine, and quantitative finance. The proliferation of applications may be attributed to *Sklar's theorem* (see appendix for details). Loosely speaking, Sklar's theorem states that any $d$-dimensional continuous joint distribution may be uniquely decomposed into $d$ marginal distribution functions and a single copula $C$. The copula precisely captures dependencies between random variables in isolation from marginals. This allows for the creation of *non-independent* distributions by combining marginals—potentially from different families and tying them together using a suitable copula.

## 2.2 Archimedean copulas

In this paper, we will restrict ourselves to *Archimedean copulas*. Archimedean copulas enjoy simplicity by modeling dependencies in high dimensions using a single 1-dimensional function:

$$C(u_1, \cdots, u_d) = \varphi \left( \varphi^{-1}(u_1) + \varphi^{-1}(u_1) + \cdots \varphi^{-1}(u_d) \right), \quad (2)$$

where $\varphi : [0, \infty) \to [0, 1]$ is $d$-monotone, i.e., $(-1)^k \varphi^{(k)}(t) \geq 0$ for all $k \leq d, t \geq 0$.

Here, $\varphi$ is known as the *generator* of $C$. A single $d$-monotone function $\varphi$ defines a $d$-dimensional copula which satisfies the conditions laid out in Section 2.1. We say that $\varphi$ is *completely monotone* if $(-1)^k \varphi^{(k)}(t) \geq 0$ for all values of $k$. Completely monotone generators define copula regardless of the dimension $d$. Most (but not all) Archimedean copula are defined by completely monotone generators. For this reason, we focus on Archimedean copula with completely monotone generators, also known in the literature as *extendible* Archimedean copula. The following theorem by Bernstein (see [27] for details) characterizes all completely monotone $\varphi$ as a mixture of exponential functions.

**Theorem 1** (Bernstein-Widder). *A generator $\varphi$ is completely monotone if and only if $\varphi$ is the Laplace transform of a positive random variable $M$, i.e., $\varphi(t) = \mathbb{E}_M(\exp(-tM))$ and $\mathbb{P}(M > 0) = 1$.*

In fact, [25] show that $C$ has an easy interpretation in terms of the random variable $M$. Specifically, if $U = (U_1, \cdots, U_d) \sim C$, where $C$ is generated by $\varphi$, which is in turn the Laplace transform of some non-negative random variable $M$ which almost never takes the value 0, then, we have $U_i = \varphi(E_i/M)$, where $E_i \sim \text{Exp}(1)$. It follows that sampling from $C$ is easy and efficient given access to a sampler for $M$ and an oracle for $\varphi$, which is the case for most commonly used copulas.

## 2.3 Related work

Copulas offer a wide range of applications, from finance and actuarial sciences [3, 1, 8, 31] to epidemiology [5, 21], engineering [32, 4] and disaster modeling [2, 24]. Copulas are popular for modeling extreme tail distributions. Recently, [35] show that GANs and Normalizing Flows suffer from inherent limitations in modeling tail dependencies and propose using copulas to explicitly do so.

In lockstep with this proliferation of applications is the introduction of more sophisticated copulas and training/learning methods. Vine copula and Copula bayesian networks [15, 16, 7] extend bivariate parametric copula to higher dimensions; the former models high dimensional distributions using a collection of bivariate copula organised in a tree-like structure, while the latter extends bayesian networks while using copulas to reparameterize conditional densities. Various mixture methods are also frequently used [29, 33, 31, 19] to construct richer representations from existing copula. Other methods include non-parametric or semiparametric methods [36, 13, 14]. In terms of model selection, [10] introduce Copula Information Criterion (CIC), an analog to classical AIC and BIC methods for copula.

In the domain of deep neural networks, popular generative models include Generative Adversarial Networks [9], Variational Autoencoders [20], and Normalizing Flow methods [30, 6]. These methods either describe a generative process or learn densities directly, as opposed to the joint distribution function. Unless explicitly designed to do so, these models are ill suited to inference on quantities such as conditional densities or distributions, while ACNet may do so via simple operations.

## 3 Archimedean Copula networks

Bernstein's theorem states that completely monotone functions are essentially mixtures of (potentially infinitely many) negative exponentials. This suggests that generators $\varphi$ could be approximated by a *finite* sum of negative exponentials, which in turn defines an approximation for (extendible) Archimedean copula. Motivated by this, our proposed model parameterizes $\varphi$ using a large but finite mixture of negative exponentials. We achieve this large mixture (often exponential in model size) of exponentials using deep neural networks.[1] We term the resultant network *Archimedean-Copula Networks*, or ACNet for short.

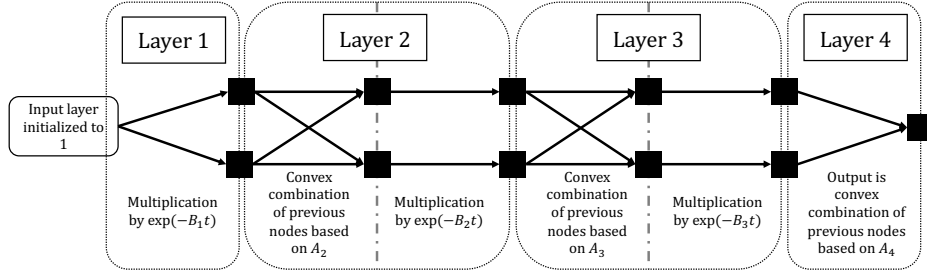

Figure 1: Forward pass through ACNet with $L = 3, H_1 = H_2 = H_3 = 2$

## 3.1 Representing $C$ from neural network representations of $\varphi$

The key component of our model is the a neural network module $\{\varphi^{\text{nn}}\} : [0, \infty) \to [0, 1]$ specifying the generator and implicitly, the copula. For simplicity we will assume that the network contains $L$ hidden layers with the $\ell$-th layer being of width $H_\ell$. For convenience, the widths of the input and output layers are written as $H_0 = 1$ and $H_{L+1} = 1$. Layer $\ell$ has outputs of size $H_\ell$, denoted by $\{\varphi^{\text{nn}}\}_{\ell,i}$ where $i \in \{1, \dots, H_\ell\}$. Structurally, $\{\varphi^{\text{nn}}\}$ looks similar to a standard feedforward network, with the additional characteristic that outputs for each layer is a convex combination of a finite number of negative exponentials (in inputs $t$). Specifically, our network has the following representation.

$$\{\varphi^{\text{nn}}\}_{0,1}(t) = 1 \qquad \text{(Input layer)}$$

$$\{\varphi^{\text{nn}}\}_{\ell,i}(t) = \exp(-B_{\ell,i} \cdot t) \sum_{j=1}^{H_{\ell-1}} A_{\ell,i,j} \{\varphi^{\text{nn}}\}_{\ell-1,j}(t) \qquad \forall \ell \in [L], i \in [H_\ell] \quad \text{(Hidden layers)}$$

$$\{\varphi^{\text{nn}}\}(t) = \{\varphi^{\text{nn}}\}_{L+1,1}(t) = \sum_{j=1}^{H_L} A_{L+1,1,j} \{\varphi^{\text{nn}}\}_{L,j}(t) \qquad \text{(Output layer)}$$

Each $A_\ell$ is a non-negative matrix of dimension $H_\ell \times H_{\ell-1}$ with each row lying on the $H_{\ell-1}$-dimension probability simplex, i.e., $\sum_{j=1}^{H_{\ell-1}} A_{\ell,i,j} = 1$. Each $B_\ell$ is a non-negative vector of size $H_\ell$. Each unit in layer $\ell$ is formed by taking a convex combination of units in the previous layer, followed by multiplying this by some negative exponential of the form $\exp(-\beta t)$, where the latter is analogous to the 'bias' term commonly found in feedforward networks. When $L = 1$, we get that $\{\varphi^{\text{nn}}\}(t)$ is equal to a convex combination of negative exponentials with rates of decay and weighting given by $B$ and $A$ respectively. A graphical representation of $\{\varphi^{\text{nn}}\}$ is shown in Figure 1.

**Theorem 2.** $\{\varphi^{nn}\}(t)$ *is a completely monotone function with domain* $[0, \infty)$ *and range* $[0, 1]$.

*Proof.* (Sketch) Since sums of exponentials are 'closed' under addition and multiplications of sums of exponentials, $\{\varphi^{\text{nn}}\}$ remains a convex combination of negative exponentials when $L > 1$. $\qquad \square$

It follows from Theorem 2 that $\{\varphi^{\text{nn}}\}$ is a valid generator for all $d \geq 2$. To ensure that $B$ is strictly positive and $A$ lies on the probability simplex, we perform the following reparameterization. Let $\Phi = \{\Phi_A, \Phi_B\}$ be the network weights underlying parameters $A$ and $B$. By setting $B = \exp(\Phi_B)$, $A_{\ell,i,j} = \text{softmax}(\Phi_{A,\ell,i})_j$ and optimizing over $\Phi$, we ensure that the required constraints are satisfied.

## 3.2 Extracting probabilistic quantities from $\{\varphi^{\text{nn}}\}$

With $\{\varphi^{\text{nn}}\}$, we are now in a position to evaluate the copula $C$ using Equation (2). This requires the computation of $\{\varphi^{\text{nn}}\}^{-1}(u_i)$, which has no simple closed form. However, we may compute this inverse efficiently using Newton's root-finding method, i.e., by solving for $t$ in the equation $\{\varphi^{\text{nn}}\}(t) - u_i = 0$. The $k$-th iteration of Newton's method involves computing the gradient $\{\varphi^{\text{nn}}\}'(t_k)$ and taking a suitable step. The gradient of $\{\varphi^{\text{nn}}\}$ is readily obtained using auto-differentiation libraries such as PyTorch [28] and typically involves a 'backward' pass through the network. Empirically, root finding typically takes fewer than 50 iterations, i.e., computing $\{\varphi^{\text{nn}}\}^{-1}(u)$ requires an effectively constant number of forward and backward passes over $\{\varphi^{\text{nn}}\}$.

## 3.3 Training ACNet by minimizing negative log-likelihood

Suppose we are given a dataset $\mathcal{D}$ of size $m$, $\{x_1, \cdots x_m\}$, where each $x_j$ is a $d$-dimensional feature suitably normalized to $[0,1]^d$. We want to fit ACNet to $\mathcal{D}$ by minimizing the negative log-likelihood $-\sum_{j=1}^m \log\left(p(x_{j_1}, \cdots, x_{j_d})\right)$ via gradient descent. The density function for a single point may be obtained by differentiating $C$ over each of its parameters once,

$$p(u_1, \cdots, u_d) = \frac{\partial^{(d)} C(u_1, \ldots, u_d)}{\partial u_1, \ldots, \partial u_d} = \frac{\varphi^{(d)}(\varphi^{-1}(u_1) + \cdots + \varphi^{-1}(u_d)))}{\prod_{i=1}^d \varphi'(\varphi^{-1}(u_i))}. \tag{3}$$

Gradient descent and backpropagation requires us to provide derivatives of $p$ with respect to the network parameters $\Phi$. This requires taking derivatives of the expression in Equation (3) with respect to $\Phi$. In general, automatic differentiation libraries such as PyTorch [28] allow for higher derivatives to be readily computed by repeated application of the chain rule. This process typically requires the user to furnish (often implicitly) the gradients of each constituent function in the expression. However, automatic differentiation libraries do not have the built-in capability to compute gradients (given $\{\varphi^{\mathrm{nn}}\}$) both with respect to inputs $u$ and network weights $\Phi$ of $\{\varphi^{\mathrm{nn}}\}^{-1}$, the latter of which is required for optimization of $\Phi$ via gradient descent.

To overcome this, we write a wrapper allowing for inverses of 1-dimensional functions to be computed via Newton's method. When given a function $\varphi(u; \Phi)$ parameterized by $\Phi$, our wrapper computes $\varphi^{-1}(u; \Phi)$ and provides the derivatives $\frac{\partial \varphi^{-1}(u;\Phi)}{\partial u}$ and $\frac{\partial \varphi^{-1}(u;\Phi)}{\partial \Phi}$. The analytical expressions for both derivatives are shown below, with derivations deferred to the appendix.

$$\frac{\partial \varphi^{-1}(u;\Phi)}{\partial u} = 1 \Big/ \frac{\partial \varphi(t;\Phi)}{\partial t} \qquad\qquad \frac{\partial \varphi^{-1}(u;\Phi)}{\partial \Phi} = -\frac{\partial \varphi(t;\Phi)}{\partial \Phi} \Big/ \frac{\partial \varphi(t;\Phi)}{\partial t}$$

Here, the derivatives are evaluated at $t = \varphi^{-1}(u; \Phi)$. By supplying these derivatives to an automatic differentiation library, $\varphi^{-1}(u; \Phi)$ can be computed in a fully differentiable fashion, allowing for computation of higher order derivatives and nested application of the chain rule to be done seamlessly. Consequently, Equation (3) and its derivatives may be easily computed without any further manual specification of gradients. Our implementation employs PyTorch [28] for automatic differentiation.

## 3.4 Interpretation of network weights

According to Bernstein's theorem (Theorem 1), $\{\varphi^{\mathrm{nn}}\}$ is the Laplace transform of some non-negative random variable $M$. Interestingly, the network structure of ACNet allows us to obtain an analytical representation of the distribution $M$. Since $\{\varphi^{\mathrm{nn}}\}$ is the sum of negative exponentials, $M$ is a discrete distribution with support given by the decay rates of $\{\varphi^{\mathrm{nn}}\}$. However, the structure of ACNet allows us to go further by implicitly describing a Markov reward model governing the mixing variable $M$.

Take the structure of ACNet as directed acyclic graph with reversed edges and consider a random walk starting from the *output*. The sampler begins with a reward of 0. The probability of transition from the $j$-th node in layer $\ell - 1$ to the $i$-th node of layer $\ell$ is $A_{\ell,i,j}$. When this occurs, it accumulates a reward of $B_{\ell,i}$. The process terminates when we reach the *input* node, where the realization of $M$ is the total reward accumulated throughout. Details can be found in the appendix.

The above interpretation has two consequences. First, the size of the support of $M$ is upper bounded by the number of possible *paths* that the Markov model possesses, which is typically exponential in $L$. This shows that deeper nets allow for distributions with an exponentially larger support of $M$ compared to shallow nets. Second, this hierarchical representation gives an efficient sampler for $M$, which can be exploited alongside the algorithm of [25] (see Section 2.2) to give an efficient sampling algorithm for $U$. More details may be found in the appendix.

## 3.5 Obtaining probabilistic quantities from ACNet

In Section 3.3, we trained ACNet by minimizing the log-loss of $\mathcal{D}$, where the likelihood $p(u_1, \ldots, u_d)$ was obtained by repeated differentiation of the copula $C$ (Equation (3)). Many other probabilistic quantities are often of interest, with applications in both inference and training.

**Scenario 1 (Inference).** Consider the setting where one utilizes surveys to study the correlation between one's age and income. Some natural inference problem follow, such as: given the age of a

respondent, how likely is it that his income lies below a certain threshold, i.e., $\mathbb{P}\left(U_1 \leq u_1 | U_2 = u_2\right)$. Similarly, one could be interested in conditional densities $p(u_1 | u_2)$ in order to facilitate conditional sampling using MCMC or for visualization purposes. We want our learned model to be able to answer *all* such queries efficiently without modifying its structure for each type of query.

**Scenario 2 (Training with uncertain data).** Now, consider a related scenario where for respondents sometimes only report the *range* of their age and incomes (e.g., age is in the range 21-25), even though underlying quantities are inherently continuous. To complicate matters, the dataset $\mathcal{D}$ is the amalgamation of multiple studies, each prescribing a different partition of ranges, i.e., $\mathcal{D}$ has rows containing a *range* of possible values for each respondent, i.e., $\left(\left(\underline{u_1}, \overline{u_1}\right), \left(\underline{u_2}, \overline{u_2}\right)\right)$, where $\underline{u_i} \leq U_i \leq \overline{u_i}$. Our goal is to learn a joint distribution which respects this 'uncertainty' in $\mathcal{D}$.[2]

To the best of our knowledge, no existing deep generative model is able to meet the demands of both scenarios. It turns out that many of these quantities may be obtained from $C$ using relatively simple operations. Suppose without loss of generality that one has observed that the first $k \in [d]$ random variables $X_K = \{X_1, \cdots, X_k\} \subseteq X$ and obtain values $x_K = (x_1, \cdots, x_k)$. We want to compute the posterior distribution of the next $d-k$ unobserved variables $X_{\bar{K}} = X \backslash X_K = \{X_{k+1}, \cdots, X_d\}$ with $x_{\bar{K}}$ analogously denoting their values. Then, the conditional distribution $\mathbb{P}(X_{\bar{K}} \leq x_{\bar{K}} | X_K = x_K)$ is the distribution function given that $X_K$ takes values $x_K$. We have the following expression

$$\mathbb{P}(X_{\bar{K}} \leq x_{\bar{K}} | X_K = x_K) = \int_{-\infty}^{x_{\bar{K}}} p(x_K, z) / p(x_K) dz = \frac{\partial F(x_K, x_{\bar{K}})}{\partial x_1 \cdots \partial x_k} \Big/ \frac{\partial F(x_K, 1)}{\partial x_1 \cdots \partial x_k},$$

where the last equality follows from $\int_{-\infty}^{x_{\bar{K}}} p(x_K, z) dz = \frac{\partial}{\partial w} \int_{-\infty}^{x_K} \int_{-\infty}^{x_{\bar{K}}} p(w, z) dw dz = \frac{\partial F(x_K, x_{\bar{K}})}{\partial x_1 \cdots \partial x_k}$. Many interesting quantities such as conditional densities $p(x_{\bar{K}} | x_K)$ may be expressed in terms of $F$ in a similar fashion, using simple arithmetic operations and differentiation. Crucially, these expressions remain differentiable and may be evaluated efficiently. Since these derivations apply for any cumulative distribution $F$, they hold for any copula $C$ as well. We list some of these commonly used probabilistic quantities and their relationship to $C$ in the appendix.

# 4 Experiments

Here, we first empirically demonstrate the efficacy of ACNet in fitting both synthetic and real-world data. We then end off by applying ACNet to Scenario 2 of Section 3.5, and show that ACNet can be used to fit data even when the data exhibits uncertainty in measurements. The goal of these experiments is *not* to serve as comparison against neural density estimators (which typically model joint *densities* and not joint *distribution* functions), but rather as an alternative to frequently used parametric copula. Experiments are conducted on a 3.1 GHz Intel Core i5 with 16 GB of RAM. We utilize the PyTorch [28] framework for automatic differentiation. We use double precision arithmetic as the inversion of $\varphi$ requires numerical precision. When using Newton's method to compute $\varphi^{-1}$, we terminate when the error is $\leq 1e - 10$. For all our experiments we use ACNet with $L = 2$ and $H_1 = H_2 = 10$, i.e., 2 hidden layers each of width 10. The network is small but sufficient for our purpose since $\{\varphi^{\text{nn}}\}$ is only 1-dimensional. $\Phi_A$ and $\Phi_B$ were initialized in the range $[0, 1]$ and $(0, 2)$ uniformly at random. We use stochastic gradient descent with a learning rate of $1e - 5$, momentum of 0.9, and a batch size of 200. No hyperparameter tuning was performed.

## 4.1 Learning known Archimedean copulas

To verify that ACNet is able to learn commonly used Archimedean copulas, we generate synthetic datasets from the Clayton, Frank and Joe copulas. These copulas exhibit different tail dependencies (see Figure 2a). For example, the Clayton copula has high lower tail-dependence but no upper-tail dependency, which makes it useful for modelling quantities such stock prices, for example, two companies involved in the same supply chain are likely to perform poorly simultaneously, but one company performing well does not imply the other will. These copula are governed by a single parameter, which are chosen to be 5, 15, and 3 respectively. For each copula, we generate 2000 train and 1000 test points and train ACNet for 40k epochs. We compare the resultant learned distribution (Figure 2b) with the ground truth (Figure 2a). Testing losses are compared in Table 5. From Figure 2

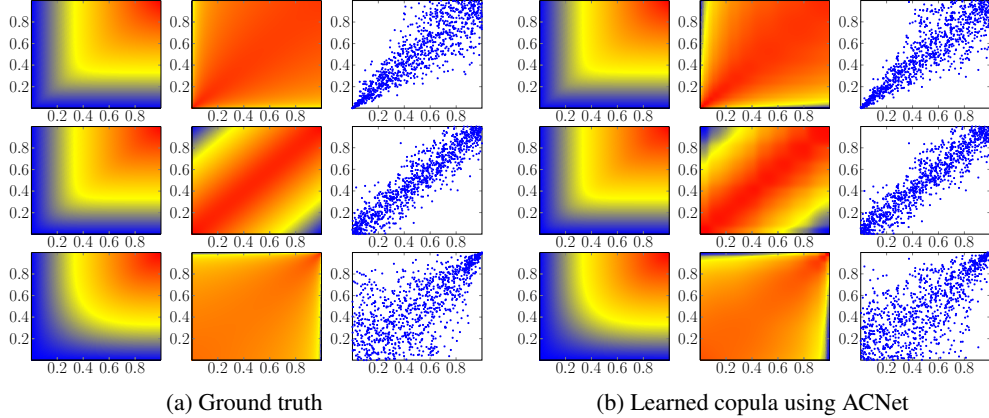

(a) Ground truth        (b) Learned copula using ACNet

Figure 2: Top to bottom: Learning Clayton, Frank and Joe copulas using ACNet. Plots from left to right: (i) joint distributions, (ii) log densities, and (iii) samples drawn from the respective distributions.

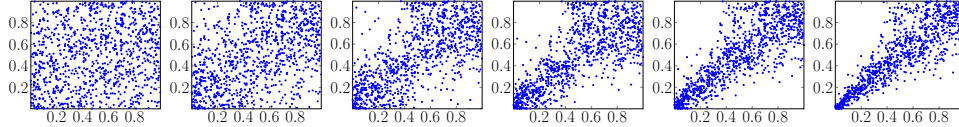

Figure 3: Left to right: Learning the Clayton copula after 0, 100, 200, 500, 1000 and 5000 epochs.

and Table 5, we can see that ACNet is able to learn all 3 copula accurately by the end of training, and the contours of the log-likelihood match the ground truth almost exactly. Figure 3 shows how the learned density changes as the number of training epochs increases for the case of the Clayton copula. We can see that as the number of training samples increases, the 'tip' at the lower tail of the copula becomes sharper, i.e., ACNet learns the lower tails of the distribution more accurately.

## 4.2 Experiments on real-world data

To demonstrate the efficacy of ACNet, we applied ACNet to 3 real-world datasets. As a preprocessing step, we normalize the data by scaling each dimension to the range $[0, 1]$ based on their ordinal ranks. This ensures that the empirical marginals are approximately uniform. Train and test sets are split based on a 3:1 ratio. We normalize both train and test sets independently. This was done to avoid leakage of information from the train to the test set, which could occur if train and test sets were normalized together. In practice, we observe no significant difference in these two methods of normalization. Because real-world data tends to contain a small number of outliers, we inject into the training set points uniformly chosen from $[0, 1]^2$. This is akin to a form of regularization and helps to prevent ACNet from overfitting. We inject 1 point for every 100 points in the training set. We repeat each experiment 5 times with different train/test splits and report the average test loss.

**Boston Housing.** We model the *negative* dependencies between per capita crime rate and the median value of owner occupied homes in Boston [11]. Since Archimedean copulas with completely monotone generators can only model positive dependencies, we insert an additional preprocessing step where we flip the data along the vertical line at $0.5$. This dataset has 506 samples.

**(INTC-MSFT)** This data comprises five years of daily log-returns (1996-2000) of Intel (INTC) and Microsoft (MSFT) stocks, and was analysed in [26]. The dataset comprises 1262 samples.

**(GOOG-FB).** We collected daily closing prices of Google (GOOG) and Facebook (FB) from May 2015 to May 2020. The data was collected using Yahoo Finance and comprises 1259 samples.

For each of the datasets, we trained ACNet based on the processed data. The learned distributions are illustrated in Figure 4. Furthermore, we compare the performance of ACNet with the Clayton, Frank and Gumbel copula and report the test log-loss of ACNet with the best fit amongst the 3 parametric

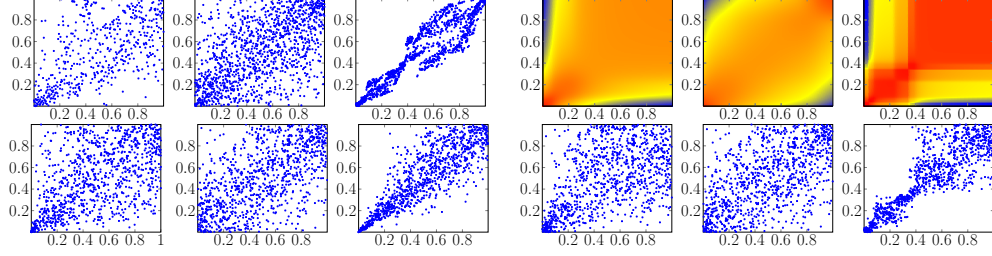

(a) Top: Data after preprocessing. Bottom: Samples (b) Learned using ACNet. Top: contour lines for log from the best-fit parametric model. densities. Bottom: Samples from learned copula.

Figure 4: Experiments for (i) Boston housing, (ii) (INTC-MSFT) and (iii) (GOOG-FB) datasets.

|         | Ground Truth | ACNet   |           |                   |         |
| ------- | ------------ | ------- | --------- | ----------------- | ------- |
|         |              |         |           | Best Parametric   | ACNet   |
| Clayton | -0.9416      | -0.9171 | Boston    | (Clayton) -0.2929 | -0.2742 |
| Joe     | -0.5111      | -0.4919 | INTC-MSFT | (Frank) -0.1947   | -0.1995 |
| Frank   | -0.8985      | -0.8759 | GOOG-FB   | (Clayton) -0.9334 | -0.9558 |

Figure 5: Testing loss over synthetic datasets. Figure 6: Testing loss over real-world datasets.

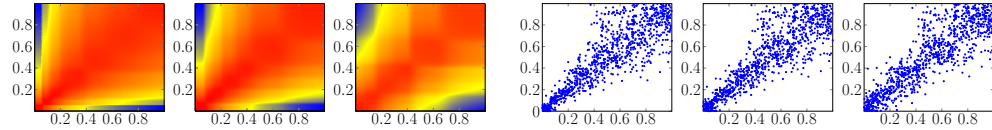

Figure 7: Learning the Clayton copula for noise parameters $\lambda = 0.1, 0.25, 0.5$ respectively. Left: Contour plots for log-densities. Right: Samples from ACNet after training.

copula (Table 6) [3]. The parametric copula were similarly trained by gradient descent.[4] Qualitatively, we observe that reasonable models were learnt for the first two datasets. For example, in the Boston housing dataset, we are able to model the higher dependence in the left tail of the distribution, and the higher testing loss is likely due to overfitting of the small dataset. In the last dataset, while ACNet is unable to exactly learn the copula, it is both *qualitatively and quantatively better* than the parametric Archimedean copulas, which are unable to model the 'two-phased' nature exhibited by this dataset.

## 4.3 Training and inference on other probabilistic quantities

Here, we demonstrate the effectiveness in applying ACNet to learning joint distributions in the presence of uncertainty in data (see Section 3.5). We use the same synthetic dataset of Section 4.1. For each datapoint, instead of observing the tuple $(u_1, u_2)$, we observe $\left(\left(\underline{u_1}, \overline{u_1}\right), \left(\underline{u_2}, \overline{u_2}\right)\right)$, where $\underline{u_i} \leq U_i \leq \overline{u_i}$. The upper and lower bounds of $u_i$ are chosen randomly such that $u_i - \underline{u_i}$ and $\overline{u_i} - u_i$ are uniformly chosen from $[0, \lambda]$, where $\lambda$ is a 'noise' parameter associated with the experiment. Note each entry has its own associated uncertainty. Fitting ACNet simply involves running gradient descent to minimize the negative log probabilities $-\log\left(\mathbb{P}\left(U_1 \in \left[\underline{u_1}, \overline{u_1}\right] \land U_d \in \left[\underline{u_2}, \overline{u_2}\right]\right)\right)$.

We experiment with $\lambda = 0.1, \lambda = 0.25, \lambda = 0.5$. Results are reported in Figure 7. In all cases, ACNet is able to learn a reasonable rendition of the Clayton copula. As expected, when $\lambda$ increases, we begin to see the inability to model the strong correlations in the lower tails. This is expected, since the uncertainty limits the degree to which we can observe strong lower tail dependencies.

## 4.4 Practical considerations and limitations of ACNet

**Experiments when** $d > 2$**.** Here, we show that ACNet is capable of fitting distributions with more than 2 dimensions. We use the GAS dataset [34], which comprises readings from chemical sensors

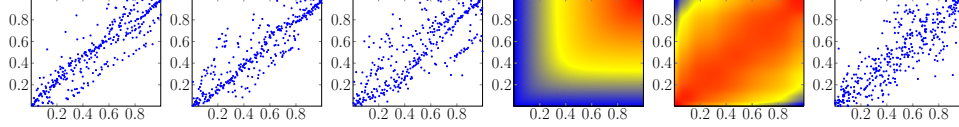

Figure 8: Left to right: (i)-(iii) Normalized training data for dimensions $(0, 1)$, $(0, 2)$ and $(1, 2)$. (iv)-(vi) joint distributions, log-densities and samples drawn from the trained network.

used in simulations for drift compensation. To simplify the situation, we use features 0, 4 and 7 from a single sensor during the second month (see [34] for details) and perform normalization for each feature in a similar fashion Section 4.2, yielding a dataset comprising 445 readings. The network architecture and train/test split are identical to Section 4.2.

As before, we train ACNet by minimizing log-loss and compare our results against the Clayton, Frank, and Gumbel copulas. The results are in Figure 8. We observe that ACNet is able to fit the data reasonably despite the data not being entirely symmetric over the 3 dimensions. ACNet achieves a test/train loss of -1.389 and -1.456, outperforming the Frank copula (the best performing parametric copula), which obtained a test/train loss of -1.356 and -1.357. Similar to the Boston housing dataset, ACNet overfits. This is unsurprising since the dataset is fairly small.

Generally, we do not recommend using ACNet with high dimensions. First, this often results in numerical issues since training ACNet by minimizing the log-loss requires differentiating the copula $d$ times. Generally, we observe that ACNet faces numerical problems for $d \geq 5$ even when employing double precision. Second, high dimensional data is rarely symmetric unless there is some underlying structure supporting this belief.

**Failure cases.** Not all datasets are well modelled by ACNet. Consider the POWER dataset [12] (Figure 9), which contains measurements for electric power consumption in a single household. For simplicity, we focus on the joint distribution of the power consumption between the kitchen and laundry room. Clearly, the POWER dataset is unlike the previous distributions, as it posesses a high level of 'discreteness'. Since there are few appliances in each room and each active appliance consumes a fixed amount of power, we would expect that each combination of active appliances would lead to a distinct profile in power consumption. As seen from Figure 9, ACNet is unable to accurately fit this distribution. It is worth noting however, that despite learning a distribution that appears qualitatively different, ACNet still achieves a test loss of -0.221, which is significantly better than the uniform distribution and slightly superior to the Clayton copula, the best fit among the copula we compared with.

**Running times.** ACNet's generator is represented by a neural network and is thus slower to train compared to single-parameter copulas. However, performing training is still feasible in practice. With our experimental setup, we are able to train 15 minibatches each of size 200 in 1 second without utilizing a GPU. Furthermore, in all our experiments, the network converges within $10 \cdot 4$ iterations. For a training set with 2000 points, ACNet converges in 3-5 hours. Computational costs are split roughly evenly

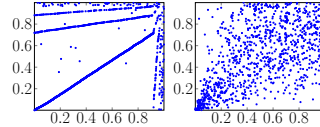

Figure 9: Left: Normalized POWER dataset. Right: Learned ACNet model.

between the forward and backward passes—the former involves solving for the inverse while the latter involves taking 2 (or more) rounds of differentiation.

## 5 Conclusion

In this paper, we propose ACNet, a novel neural network architecture which learns completely monotone generators of Archimedean copula. ACNet's network weights can be interpreted as parameters of a Markov reward process, leading to an efficient sampling algorithm. Using ACNet, one is able to compute numerous probabilistic quantities, unlike existing deep models. Empirically, ACNet is able to match or outperform common Archimedean copulas in fitting synthetic and real-world data, and is also able to learn in the presence of uncertainty in data. Future work include moving beyond completely monotone generators, learning hierarchical Archimedean copulas, as well as developing methods to jointly learn marginals.

# 6 Broader impact statement

Copulas have held the dubious honor of being partially responsible for the financial crisis of 2008 [23]. Back then, it was commonplace for analysts and traders to model prices of collateralized debt obligations (CDOs) by means of the Gaussian copula [22]. Gaussian copulas were extremely simple and gained popularity rapidly. Yet today, this method is widely criticised as being overly simplistic as it effectively summarizes associations between securities into a single number. Of course, copulas now have found a much wider range of applications, many of which are more grounded than credit and risk modeling. Nonetheless, the criticism that Gaussian—or for that matter, any simple parametric measure of dependency is too simple, still stands.

ACNet is one attempt to tackle this problem, possibly beyond financial applications. While still retaining the theoretical properties of Archimedean copula, ACNet can model dependencies which have no simple parametric form, and can alleviate some difficulties researchers have when facing the problem of model selection. We hope that with a more complex model, the use of ACNet will be able to overcome some of the deficiencies exhibited by Gaussian copula. Nonetheless, we continue to stress caution in the careless or flagrant application of copulas—or the overreliance on probabilistic modeling—in domains where such assumptions are not grounded.

At a level closer to machine learning, ACNet essentially models (a restricted set of) cumulative distributions. As described in the paper, this has various applications (see for example, Scenario 2 in Section 3 of our paper), since it is computationally easy to obtain (conditional) densities from the distribution function, but not the other way round. We hope that ACNet will motivate researchers to explore alternatives to learning density functions and apply them where appropriate.

# 7 Funding transparency statement

Co-authors Ling and Fang are supported in part by a research grant from Lockheed Martin. The views and conclusions contained in this document are those of the authors and should not be interpreted as representing the official policies, either expressed or implied, of Lockheed Martin.

## Footnotes

[1]Approximating completely monotone functions using sums of exponentials has been studied [18, 17], but not in the context required for learning copula.

[2]Unlike usual settings, we are not adding or assuming a known noise distribution but rather, assume that our data is known to a lower precision.

[3]We report the best performing model, with and without regularization.

[4]There are multiple ways of training parametric copula—for example, by matching concordance measures such as Kendall's Tau and Spearman's Rho. We do not consider these alternative fitting methods here.

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
