[Supplementary Material]

# 8  Appendix

## 8.1  Sklar's Theorem

**Theorem 3** (Sklar, 1959). *Let $F$ be a distribution function with margins $F_1, \ldots F_d$. Then there exists a d-dimensional copula $C$ such that for all $(x_1, \ldots, x_d) \in \mathbb{R}^d$) it holds that $F(x_1, \ldots, x_d) = C(F(x_1), \ldots, F(x_d))$. Furthermore, if $F_1, \ldots, F_d$ are continuous, then $C$ is unique. Conversely, if $C$ is a d-dimensional copula and $F_1, \ldots, F_d$ are univariate distribution functions, then $F(x_1, \ldots, x_d) = C(F(x_1), \ldots, F(x_d))$ is a d-dimensional distribution.*

## 8.2  Derivations for deratives of inverses

If $g$ is the inverse of $f$, that is, $g_w(y) = f_w^{-1}(y)$ or $g_w(f_w(t)) = t$ for some weights $w$. If we treat $w$ as parameters as well, then we have scalar functions $g(a, b)$ and $f(c, d)$ such that the identity

$$g(f(t, w), w) = t$$

holds for all possible $w$.

**Part 1.**  We want to find $\left.\frac{\partial g(y,r)}{\partial y}\right|_{\substack{y=a \\ r=w}}$. Since $f$ and $g$ are scalar functions of $y$, it is easy to see geometrically that

$$\left.\frac{\partial g(y,r)}{\partial y}\right|_{\substack{y=a \\ r=w}} = 1 \left/ \left(\left.\frac{\partial f(x,r)}{\partial x}\right|_{\substack{x=g(a,w) \\ r=w}}\right)\right.$$

**Part 2.**  We want to find $\left.\frac{\partial g(y,r)}{\partial r}\right|_{\substack{y=a \\ r=w}}$ for a given $w$ and $a$, given access to an oracle $f(x,r)$, $g(y,r)$, $\frac{\partial f(x,r)}{\partial r}$, $\frac{\partial f(x,r)}{\partial x}$ and for any values of $x, y, r$. Here, evaluating $g(y, w)$ requires a call to Newton's method and the 2 partial derivatives may be obtained from autograd. Taking *full* derivatives of the identity $g(f(t, w), w) = t$ with respect to $w$ yields

$$\frac{dg(f(t,w),w)}{dw} = \frac{\partial g}{\partial f}\frac{\partial f}{\partial w} + \frac{\partial g}{\partial w}$$

$$= \left(\left.\frac{\partial g(y,r)}{\partial y}\right|_{\substack{y=f(t,w) \\ r=w}}\right) \cdot \left(\left.\frac{\partial f(x,r)}{\partial r}\right|_{\substack{x=t \\ r=w}}\right) + \left.\frac{\partial g(y,r)}{\partial r}\right|_{\substack{y=f(t,w) \\ r=w}}$$

$$= 0$$

$$\left.\frac{\partial g(y,r)}{\partial r}\right|_{\substack{y=f(t,w) \\ r=w}} = -\left(\left.\frac{\partial g(y,r)}{\partial y}\right|_{\substack{y=f(t,w) \\ r=w}}\right) \cdot \left(\left.\frac{\partial f(x,r)}{\partial r}\right|_{\substack{x=t \\ r=w}}\right)$$

Note that this holds for all $t$. Performing a substitution gives

$$\left.\frac{\partial g(y,r)}{\partial r}\right|_{\substack{y=a \\ r=w}} = -\left(\left.\frac{\partial g(y,r)}{\partial y}\right|_{\substack{y=a \\ r=w}}\right) \cdot \left(\left.\frac{\partial f(x,r)}{\partial r}\right|_{\substack{x=g(a,w) \\ r=w}}\right)$$

$$= -\left(\left.\frac{\partial f(x,r)}{\partial r}\right|_{\substack{x=g(a,w) \\ r=w}}\right) \left/ \left(\left.\frac{\partial f(x,r)}{\partial x}\right|_{\substack{x=g(a,w) \\ r=w}}\right)\right. ,$$

where the last line holds using $\left[h^{-1}\right]'(x) = 1/\left[h'(h^{-1}(x))\right]$ for scalar $h$ (Part 1).

## 8.3  Proof of Theorem 2

We first show that the output at each layer $\{\varphi^{\text{nn}}\}(t)$ is a convex combination of negative exponentials, i.e.,

$$\{\varphi^{\text{nn}}\}_{\ell,i}(t) = \sum_{k=1}^{K_{\ell,i}} \alpha_k \exp(-\beta_{\ell,i,k}t) \qquad \text{where} \sum_{k=1}^{K_{\ell,i}} \alpha_{\ell,i,k} = 1,$$

where $K_\ell = \prod_{q=1}^{\ell-1} H_q$ and denotes the number of components in the mixture of exponentials (with potential repetitions). The theorem is shown by induction on the layer index $\ell$. The base case when $\ell = 0$ is obvious by setting $K_{0,1} = 1, \alpha_{0,1} = 1, \beta_{0,1} = 0$. Now suppose that the induction hypothesis is true for all $\{\varphi^{\text{nn}}\}_{\ell-1,i}$, we have,

$$\{\varphi^{\text{nn}}\}_{\ell,i}(t) = \exp(-B_{\ell,i} \cdot t) \sum_{j=1}^{H_{\ell-1}} A_{\ell,i,j} \{\varphi^{\text{nn}}\}_{\ell-1,j}(t)$$

$$= \exp(-B_{\ell,i} \cdot t) \sum_{j=1}^{H_{\ell-1}} A_{\ell,i,j} \sum_{k=1}^{K_{\ell-1}} \alpha_{\ell-1,j,k} \exp(-\beta_{\ell-1,j,k}t) \qquad \text{(induction hypothesis)}$$

$$= \sum_{j=1}^{H_{\ell-1}} \sum_{k=1}^{K_{\ell-1}} \underbrace{A_{\ell,i,j}\alpha_{\ell-1,j,k}}_{\alpha_{\ell,i,\cdot}} \exp(-\underbrace{(\beta_{\ell-1,j,k} + B_{\ell,i})}_{\beta_{\ell,i,\cdot}} t)$$

$$= \sum_{k=1}^{K_\ell} \alpha_{\ell,i,k} \exp(-\beta_{\ell,i,k}t). \qquad (4)$$

In the third and fourth line, we can also see that $\sum_{k=1}^{K_\ell} \alpha_{\ell,i,k}$ since from the induction hypothesis $\sum_{k=1}^{K_{\ell-1}} \alpha_{\ell-1,j,k} = 1$ and the design of ACNet, which guarantees $\sum_{j=1}^{H_{\ell-1}} A_{\ell,i,j} = 1$. Theorem 2 follows from the fact that sum of completely monotone functons are also completely monotone. The range of $\{\varphi^{\text{nn}}\}$ follows directly from it being a convex combination of negative exponentials.

## 8.4 Representation of $M$ in ACNet as a Markov reward process

It is known that Archimedean copula with completely monotone generators are *extendible*, and have generators $\varphi$ which are Laplace transforms of (almost surely) positive random variables $M$. The random variable $M$ is known as the *mixing variable* in a manner analogous to the De Finetti's theorem (observe that Archimedean copula are exchangable), such that a sample from the copula $C$ is given by $(\varphi(E_1/M), \ldots, \varphi(E_d/M))$, where the $E_i$ are i.i.d. samples from an exponential distribution with scale parameter 1. Hence, $M$ is known as the mixing(latent) variable, since each $U_i$ is independent of $U_j, i \neq j$ conditioned on $M$. For more information about extendible copula, refer to Chapters 1-3 of Matthias, Scherer, and Mai Jan-frederik.

From the derivations in (4), it can be seen that for all $\ell \in [L], i \in [H_\ell], k \in [K_{\ell,i}]$, we have

$$\beta_{\ell,i,k} = \sum_{q=1}^{\ell} B_{\ell,z_q^k}, \qquad \alpha_{\ell,i} = \prod_{\ell'=1}^{\ell} A_{\ell',z_{\ell'}^k, z_{\ell'-1}^k}$$

where $z_q \in [H_q]$ such that the sequence of nodes $((0, z_0^k = 1), (1, z_1^k), \ldots, (\ell-1, z_{\ell-1}^k), (\ell, z_i^k))$, each given of the form (layer, index), represents a forward path along the directed acyclic graph prescribed by the layers of the network, starting from the input node to the node $(\ell, i)$. For the $i$-th output in the $\ell$-th layer, each constituent decay weight $\beta_{\ell,i,k}$ is the sum of '$B$-terms' taken along some path starting from the input node and ending at the $(\ell, i)$-th node. Similarly, the $\alpha_{\ell,i,k}$ terms are the *product* of weights of convex combinations, given by the '$A$-terms' taken along that same path. Each term in the summand of (4) has a one-to-one mapping with such a path.

Consequently, each constituent exponential function in the output node is represented by a path $((0, z_0), (1, z_1), \ldots, (L, z_L), (L+1, 1))$. Let $\mathcal{P}$ be the set of all such paths, where the $k$-th path is given by $p_k = ((0, z_0^k) = 1, (1, z_1^k), \ldots, (L, z_L^k), (L+1, z_{L+1}^k = 1))$.

$$\{\varphi^{\text{nn}}\}_{L+1,1}(t) = \sum_{k=1}^{K_{L+1}} \alpha_{L+1,1,k} \exp(-\beta_{\ell,i,k}t)$$

$$= \sum_{p_k \in \mathcal{P}} \left( \prod_{\ell=1}^{L+1} A_{\ell,z_\ell^k, z_{\ell-1}^k} \right) \left( \exp(-(\sum_{\ell=1}^{L} B_{\ell,z_\ell^k})t) \right)$$

$$= \mathcal{L} \left\{ \sum_{p_k \in \mathcal{P}} \left( \prod_{\ell=1}^{L+1} A_{\ell,z_\ell^k, z_{\ell-1}^k} \right) \delta \left( t - \sum_{\ell=1}^{L} B_{\ell,z_\ell^k} \right) \right\} \qquad (5)$$

Using the fact that $\sum_{j=1}^{H_{\ell-1}} A_{\ell,i,j} = 1$ (by the design of ACNet), we can see that each $A_\ell$ is a transition matrix from one layer to the one which *precedes* it. Since $\ell \in [L]$, $\sum_{k=1}^{K_{\ell,i}} \alpha_{\ell,i,k} = 1$, the expression in (5) is the Laplace transform of a discrete random variable $M$ taking values at $\sum_{\ell=1}^{L} B_{\ell,z_\ell^k}$ with probability $\left( \prod_{\ell=1}^{L+1} A_{\ell,z_\ell^k, z_{\ell-1}^k} \right)$,

Figure 10: Sampling $M$ starting from the output node. Labels on edges denote probabilities of transition. Numbers in boxes correspond to rewards accumulated at each hidden node. Straight lines show a potential sample path in sampling, with total ward $B_{1,1} + B_{2,1}$.

---

**Algorithm 1:** Sampling from ACNet

---

**Result:** $d$ dimensional sample from ACNet
$M \leftarrow 0$, state $\leftarrow$ output node;
**while** *state is not in first layer* **do**
  Sample next state propotionate to $A$;
  state $\leftarrow$ next state;
  Accumulate $M$ according to state based on $B$;
**end**
Draw $d$ i.i.d. samples $E_i \sim \text{Exp}(1)$ ;
**return** $(\{\varphi^{\text{nn}}\}(E_1/M), \ldots, \{\varphi^{\text{nn}}\}(E_d/M))$

---

for each possible $p_k \in \mathcal{P}$. This is precisely the random variable coressponding to the Markov reward process in the 'reversed network' with rewards $\{B_\ell\}$ and transition matrixes $\{A_\ell\}$—most notably, the transitions given by $A_\ell$ are independent of the previous transitions taken and only depend on current state. A graphical representation of this when $L = 2$ and $H_\ell = 2$ is given in Figure 10. This Markovian property is precisely why ACNet is able to represent a generator comprising an exponential (in terms of parameters) of negative exponential components. Since we can sample from $M$, we are also able to sample from the copula efficiently using the algorithm of [25]. The psuedocode for doing so is given in Algorithm 1.

## 8.5 Representational limits of ACNet

Copulas are sometimes used to model upper and lower tail-dependencies. When $d = 2$, they are quantified respectively by,

$$UTD_C = \lim_{u \to 1^-} \frac{C(u,u) - 2u + 1}{1 - u} = \lim_{u \to 1^-} \mathbb{P}(U_1 > u | U_2 > u) \qquad \text{(Upper tail dependency)}$$

$$LTD_C = \lim_{u \to 0^+} \frac{C(u,u)}{u} = \lim_{u \to 0^+} \mathbb{P}(U_1 \leq u | U_2 \leq u) \qquad \text{(Lower tail dependency)}$$

assuming those limits exist. These quantities describe the limiting dependencies in the tails of the joint distribution. Many common Archimedean copula are have asymmetric tail dependencies, i.e., $UTD_C \neq LTD_C$. Both $UTD_C$ and $LTD_C$ of an Archimedean copula are closely linked to the mixing variable $M$. In particular, if $\mathbb{E}(M) < \infty$ then $UTD_C = 0$. Similarly, if $M$ is bounded away from zero, i.e., there exists $\epsilon$ such that $\mathbb{P}(M \in [0, \epsilon]) = 0$, then $LTD_C = 0$. Since $M$ is discrete with a finite support, both these conditions are satisfied and $UTD_C$ and $LTD_C$ are equal to 0.

## 8.6 Probabilistic quantities derivable from $C$ (or $F$)

Table 1 gives a list of some of the common probabilistic quantities which can be derived from $C$ (or $F$).

| Name | Expression | Formula in terms of $C$ or $F$ |
|---|---|---|
| Distribution | $C(u_1, \ldots, u_d)$ | $C(u_1, \ldots, u_d)$ |
| Likelihood | $p(u_1, \ldots u_d)$ | $\frac{\partial^d C(u_1, \ldots, u_d)}{\partial u_1, \ldots, \partial u_d}$ |
| Cond. Distribution | $\mathbb{P}(X_{\bar{K}} \leq x_{\bar{K}} \mid X_K = x_K)$ | $\frac{\partial F(x_K, x_{\bar{K}})}{\partial x_1 \cdots \partial x_k} \Big/ \frac{\partial F(x_K, 1)}{\partial x_1, \cdots, \partial x_k}$ |
| Cond. Likelihood | $p(X_{\bar{K}} = x_{\bar{K}} \mid X_K = x_K)$ | $\frac{\partial F(x_K, x_{\bar{K}})}{\partial x_1 \cdots \partial x_d} \Big/ \frac{\partial F(x_K, 1)}{\partial x_1, \cdots, \partial x_k}$ |
| Probability | $\mathbb{P}\left(U_1 \in \left[\underline{u_1}, \overline{u_1}\right] \wedge \cdots \wedge U_d \in \left[\underline{u_d}, \overline{u_d}\right]\right)$ | See $d$-increasing property, (1) |

Table 1: Probabilistic quantities written in terms of derivatives of $C$ or $F$.

## 8.7 Datasets

The POWER and GAS datasets are obtained from the UCI machine learning repository (`https://archive.ics.uci.edu/ml/index.php`). The Boston housing dataset is commonly found and may be downloaded through scikit-learn (`https://scikit-learn.org/stable/datasets/index.html`) or Kaggle (`https://www.kaggle.com/c/boston-housing`). The INTC-MSFT dataset is standard in copula libraries for R (`https://rdrr.io/cran/copula/man/rdj.html`). The GOOG-FB dataset was obtained by the authors from Yahoo Finance. We will provide instructions on how to obtain the final 2 datasets alongside our source code.