[Reviews · NeurIPS 2020]

Review 1

Summary and Contributions: This paper proposes ACNet, a generic form of Archimedean copulas. The authors thoroughly study their theoretical properties. In particular they show that Archimedean copulas can be learned in noisy settings and that conditionals can be extracted easily after parameters learning. Experimentally, they show that ACNets are able to express Clayton, Joe and Frank copulas. In addition, they show that ACNet performs as well as the best parametric copula on 3 small datasets.

Strengths: The paper is well written (except typos) and easy to follow. ACNet is a novel idea, it is well introduced and the discussions about the interpretation of its parameters is very interesting. In addition, the authors show why Archimedean copulas could be preferred to other density estimation methods such as GANs, VAEs or NFs.

Weaknesses: The experiments presented in the paper are not very convincing if the authors want to keep the comparison with neural density estimation methods. Indeed they do not compare their results with NFs whereas they could easily look at the results of some standard flows on the table 2. They could also apply their method to the standard benchmarks in neural density estimation(POWER, GAS, MINIBOONE, HEPMASS and BSDS300). However, it seems to me that ACNet may be too expensive to run on large datasets due to the inversion required at both training and testing times. This does not completely destroy the contribution but this should be discussed if this is something that avoids further comparison. Finally, the comparison with the parametric copulas show that ACNet do not lead to improvement if we are able to select the best parametric copula and thus again comes the computation time question: Is training ACNet faster than training the 3 types of copulas considered in this work?

Correctness: The empirical methodology could be improved. I suggest the following: - Error bars. - Explicit test and train split (do you perform the uniformisation independently?). - Computing time. - Comparison to other methods and on more standard benchmarks.

Clarity: The paper is well written but there are many singular/plural mistakes in the text.

Relation to Prior Work: I think authors should perform a thorougher comparison with other density estimation techniques (flows + other copulas + neural nets related works). You could take a look at the following papers: - Masked autoregressive flows for density estimation - Neural autoregressive flows - Neural spline flows.

Reproducibility: Yes

Additional Feedback: ——————— Post review response feedback ———————— In light of the other reviews and the response sent by the authors I believe the idea presented is sound and deserved to be publicized and I decided to raise my score to 6. However the authors have a lot of work to achieve in order to provide the results they claim they will add. What intruigues me most is the response regarding the way the authors performed uniformisation. The authors said they performed it independently on the train and test. It seems to me that this is in general a very bad practice. Instead you should consider each test sample independently and thus be able to apply a uniformisation even if the test set is made of few samples. I would like the authors to clarify what they did in their experiments and be careful with such uncommon (because of undesired) practice. In addition the authors avoided to comment on my question regarding the computation cost compared to the copulas they compare to. This is not yet clear to me what exactly is the advantage of using the proposed method instead of testing the other copulas, this should probably be clarified somewhere.


Review 2

Summary and Contributions: This paper proposes a differentiable way to estimate Archimedean Copulas. This is especially interesting to approximate Copulas which can't be estimated in closed form. This enables learning such Copulas but at the same time benefits of the benefits offered by the Copula approach. In their results they show that it performs similar or better than Clayton, Frank and Gumbel Copulas on real and synthetic data. I think the concerns other reviewers are raising are valid. And I also agree that they are not all properly addressed by the authors. I feel like the authors included ~50% of the feedback of the reviewers to improve the manuscript and are defensive on the other half (which is not that uncommon). However I still think the paper could be accepted and mainly because of two reasons: - we kind of all agree that the paper is interesting and unique. I think the paper could make the conference more interesting and diversify the topics of accepted papers. Also no of the original reviews was worse than marginally below threshold - we probably all agree that the method might be limited in certain directions (mainly data set complexity or computational). I personally don't value computational times that high (especially if all of them are under 2 hours) - the real issues usually arise when the computational complexity explodes which I don't think is the case here.

Strengths: The paper is nicely written and motivated. The motivation to use Copulas is convincing and the approach taken is sound to me. The experiments show the basic properties necessary to render the approach useful.

Weaknesses: They did not cite my work, when talking about where Copulas are applied :) I guess the only thing that I would see as negative is probably the experimental evaluation. Whilst the evaluation done is good and necessary, it might not convince everybody. Especially it would be nice to compare against Copula networks or the mixture methods cited in the related works section. I'm however not familiar with those works, and am therefore not sure if this comparison would be feasible and fair?

Correctness: The methods are described and formulated nicely. I however did not check all the details carefully, especially not the sketch of the proof of theorem 2

Clarity: The paper is nicely written

Relation to Prior Work: yes - except for not citing my work :P

Reproducibility: Yes

Additional Feedback: The broader impact statement was a joy to read - at the same time it is legit.


Review 3

Summary and Contributions: The paper introduces a method for learning Archimedean Copulas by making use of deep learning. Specifically, the introduce a novel differentiable neural network architecture to this end.

Strengths: Archimedean copulas are an important tool for data-driven learning of the dependence structure between variables. The idea of using a deep learning architecture to learn the copula is novel and useful. The proposed architecture is correctly crafted to allow for effecting the copula learning problem. Its usefulness becomes more immense when it comes to modelling dependencies between more than two variables. Traditionally, this is a problem difficult to address; for instance, vines structures are used, which are both cumbersome to design and hard to effectively train.

Weaknesses: The provided empirical evidence is far from convincing. We needed many more datasets to compare with, and especially comparisons against alternative vines in the context of multivariate scenarios. A comparative discussion on computational costs is also a must.

Correctness: The theoretical claims are correct, as is the derivation of the model.

Clarity: The paper is generally well-written, and provides all the details needed for it to be self-contained.

Relation to Prior Work: The authors provide a brief summary on related work. I do have the feeling, though, that a bit lengthier discussion on vines would help the reader unfamiliar with the topic.

Reproducibility: Yes

Additional Feedback: I have read the rebuttal. It clarified some issues, but also left many points that need further consideration.


Review 4

Summary and Contributions: This paper proposed a new neural network module for estimating the copula of data within the Archimedean copula family. In particular, the generator characterizing this Archimedean copula is parameterized with the proposed ACNet structure, which is designed to satisfy the completely monotone property and differentiable. In addition, a Markov renewal interpretation on the weights is provided. Empirical results show the proposed approach is flexible enough to fit data generated from known Archimedean copulas. For real data, the proposed approach yields even lower test loss than the parametric copulas considered.

Strengths: The proposed approach is novel in terms of designing a new neural network architecture to learn the generator of copula, improving the tradeoff between flexibility and tractability within the Archimedean copula family. The copula-based framework focuses on the dependence structure of data and neglects the marginal information. Probabilistic quantities such as conditional probabilities can be evaluated by the CDF/copula function. Despite that the Archimedean copula family has its own limitations, the experiments highlight its advantages in modeling some special aspects of data, such as tail dependencies.

Weaknesses: Although the methodology development is quite novel, the empirical validation is weak. (1) Although Archimedean copula is able to model dependence structure in high dimension, it seems in this paper only bivariate cases are evaluated. (2) Meanwhile, I am wondering whether there are other approaches for parametrizing the generator of Archimedean copula in the literature, which can serve as baselines for comparison. (3) Most of the experimental results are qualitative. How is the test loss defined? Is there a more rigorous metric for goodness-of-fit test of copulas in bivariate/high-dimension?

Correctness: The proposed approach seems correct but I didn't closely check the mathematical derivations.

Clarity: This paper is well-written and organized.

Relation to Prior Work: It seems there is a literature on semiparametric Archimedean copulas that is missing from the discussion. For example, Hernández-Lobato, José Miguel, and Alberto Suárez. "Semiparametric bivariate Archimedean copulas." Computational statistics & data analysis 55.6 (2011): 2038-2058. APA Vandenhende, François, and Philippe Lambert. "Local dependence estimation using semiparametric Archimedean copulas." Canadian Journal of Statistics 33.3 (2005): 377-388. Hoyos-Argüelles, Ricardo, and Luis Nieto-Barajas. "A Bayesian semiparametric Archimedean copula." Journal of Statistical Planning and Inference 206 (2020): 298-311. Najjari, Vadoud, Tomáš Bacigál, and Hasan Bal. "An Archimedean copula family with hyperbolic cotangent generator." International Journal of Uncertainty, Fuzziness and Knowledge-Based Systems 22.05 (2014): 761-768.

Reproducibility: Yes

Additional Feedback: What do you mean by "two-phased" nature in the dataset? ******* After Rebuttal: Thank you for the response! The points related to semi-parametric Archimedean copula approaches are fair. I still have hesitation on what makes ACNet an approach of wide applicability. I understand bivariate copula is the cornerstone for high-dimension but for the bivariate case there're also many other choices. To show the advantages of the proposed Archimedean copula-based approach I think it would be helpful to evaluate and also compare to fewer choices in high-dimensional cases (such as Gaussian copula). I also find the training with uncertainty part very interesting, but the empirical evaluation of this part needs more work.

[Author Response · NeurIPS 2020]

We thank all reviewers for their time and comments. Below, we try to address some of the main questions raised by each reviewer, and will edit the paper to include these additions.

**Reviewer 1**

1. **Comparison of ACNet with other neural density estimators.** We consider ACNet to have significantly different goals from flows and other density estimation techniques. ACNet learns Copulas rather than joint densities. It allows us to compute and backpropagate through quantities such as probabilities, conditional cumulants etc., on top of densities (Sec. 3.2 and Appx. 7.6). This has numerous applications such as the uncertain data setting we experiment on (Sec. 4.3). In contrast, common neural density estimators require numerical integration to compute these quantities and may face significant numerical challenges especially when backpropagating through them. *ACNet should be considered as a neural method for learning Copula/joint CDFs, rather than an alternative for flows and other neural density estimators.*

2. **Computational cost.** Inversion typically requires no more than 50 forward-backward passes of ACNet (Sec. 3.1). This can be regarded as a 'constant factor', though its effect is clearly non-negligible. Training is naturally slower than the parametric copula we compare to, since ACNet possesses many more parameters. We do want to point out the following. (a) Our method is able to learn Copula which parametric copula were unable to learn (closing prices of GOOG-FB), and as such, should not be simply treated as a basket of existing parametric copula. (b) In all experiments, ACNet took no more than 2 hours with a standard consumer laptop (Sec. 4). (c) The number of inverse calculations required grows linearly with the number of dimensions—this follows from the fact that the inversions are of 1-dimensional functions. (d) Inversions are done by performing Newton's method in a 'vectorized' manner. This results in very significant speedups in practice.

3. **Evaluation of ACNet using existing benchmarks for density estimation.** We choose finance as a target domain since this has been the traditional application of Copulas. Nonetheless, we agree that it would be good to apply them to the simple datasets such as GAS and POWER and will include it in the final version of the paper.

4. **Experimental setup.** The normalization of the data was done separately for train and test sets—this was done to avoid 'leakage' of information from train to test sets.

**Reviewer 2.** Thank you! If you want to anonymously suggest an additional reference, we'd be happy to add that :-).

**Reviewer 3**

1. **Empirical evaluation.** We have compared our method on both synthetic and real-world datasets, as well as a less-known setting involving noisy data. These experiments cover most common tail dependencies.

2. **Comparison to Vines.** We do not see ACNet as a competing method to Vines, but rather, a complement which augments the space of bivariate distributions which are available. ACNet, however, is an alternative towards manually selecting bivariate Archimedean Copula, which we demonstrate in our experiments. Furthermore, we believe that the topic of Vines, while interesting, is worth consideration separate from ACNet.

3. **Computational cost.** See point 2 in our response to Reviewer 1.

**Reviewer 4**

1. **Evaluating ACNet on higher dimensions.** Bivariate distributions are the cornerstone of Copula research. For example, they are the building blocks for Vines, which are a common way of scaling up Copulas. Furthermore, since Archimedean Copulas are symmetric, learning the generator for a 2-dimensional distribution would implicitly learn the generator for a n-dimensional one, assuming the assumption of symmetry holds. It is possible to evaluate ACNet on dimensions $d > 2$ (see point 2 in our response to Reviewer 1) and we will include it in the final paper.

2. **Semiparametric models.** Semi-parameteric approaches often suffer from various limitations such as slow computation and are fundamentally quite different from the neural models we consider. Nonetheless, we agree with the author and will include them in the discussion on background work.

3. **Quantitative evaluation and metrics.** As mentioned in Section 4, we have reported the testing log-likelihood lost in our experiments. This is the most common metric in the machine learning community. The qualitative evaluation is meant to supplement the quantitative evaluation. The reviewer brings up a fair point about other useful metrics in evaluating copula, such as Kendall's Tau and upper/lower tail dependencies (see Appendix 7.5). However these quantities are either difficult to compute (except in very specific parametric models) or focus on goodness of fit of tails as opposed to the whole joint distribution, which is what ACNet is designed for. Nonetheless, we will include a more thorough evaluation in the final version.

4. **Two-phased nature.** The phrase 'two-phased' was used to describe phenomena in normalized data where there appears to be distinct phases in the joint distribution—before and after 0.5, where the degree of positive dependence increases and decreases within each phase. We will make this clear in the final version of the paper.

[Meta-Review · NeurIPS 2020]

This paper was considered borderline by the reviewers. They had concerns over how the test data was treated differently from the training data and whether the experimental evaluation makes a case for the usefulness of the idea. On the plus side, the paper is very cleanly presented and the basic idea is a good on. It is currently rated borderline.